# How to Prevent Loss of Muscle Mass and Strength among Older People in Neuro-Rehabilitation?

**DOI:** 10.3390/nu11040881

**Published:** 2019-04-19

**Authors:** Aurélien Lathuilière, Julie Mareschal, Christophe E. Graf

**Affiliations:** 1Department of Rehabilitation and Geriatrics, Geneva University Hospital, 1205 Geneva, Switzerland; christophe.graf@hcuge.ch; 2Clinical Nutrition, Geneva University Hospital, 1205 Geneva, Switzerland; Julie.Mareschal@hcuge.ch

**Keywords:** stroke, muscle atrophy, sarcopenia, rehabilitation, elderly

## Abstract

Stroke is the second leading cause of death worldwide but also of disability. Stroke induces certain alterations of muscle metabolism associated with gross muscle atrophy and a decrease in muscle function, leading to sarcopenia. The vast majority of stroke cases occur in adults over 65 years of age, and the prevalence is expected to massively increase in the coming years in this population. Sarcopenia is associated with higher mortality and functional decline. Therefore, the identification of interventions that prevent muscle alterations after stroke is of great interest. The purpose of this review is to carry out a systematic literature review to identify evidence for nutritional and pharmacological interventions, which may prevent loss of muscle mass in the elderly after stroke. The search was performed on Medline in December 2018. Randomized controlled studies, observational studies and case reports conducted in the last 20 years on post-stroke patients aged 65 or older were included. In total, 684 studies were screened, and eight randomized control trials and two cohort studies were finally included and examined. This review reveals that interventions such as amino acid supplementation or anabolic steroid administration are efficient to prevent muscle mass. Little evidence is reported on nutritional aspects specifically in sarcopenia prevention after stroke. It pinpoints the need for future studies in this particular population.

## 1. Introduction

Stroke is the second leading cause of disability in the world. The number of stroke survivors with residual disabilities is constantly growing [1]. Two-thirds of hospitalized stroke cases are 65 years or more [2]. In the older patient population, the stroke outcomes are poorer with a higher mortality rate and higher functional decline [3]. Moreover, age is considered as a negative predictive factor of the rehabilitation outcome [4,5]. The factors influencing poor stroke outcome in older patients may include the higher number of comorbidities and poor nutritional status [6,7]. 

Sarcopenia classically refers to an age-related condition and typically incorporates the concept of loss of muscle mass and function [8]. It is also recognized as a type of malnutrition by the European Society of Clinical Nutrition and Metabolism (ESPEN) [9]. Sarcopenia is recognized as a geriatric syndrome, which is associated with higher mortality, higher functional decline, an increased rate of falls and a higher incidence of hospitalizations [10]. The most widely recognized etiologic factors include genetic, hormonal changes (insulin resistance), nutritional status, reduced physical activity, immobilization and inflammation [11,12]. 

The concept of stroke-related sarcopenia has emerged less than 10 years ago. This entity is supported by the existence of adaptive muscle changes after stroke and the high prevalence of malnutrition in stroke patients [13]. In stroke patients, malnutrition or dysphagia occur in up to 49% or 52.6%, respectively [14]. Stroke can directly induce neurogenic dysphagia through sensory or motor impairment in the swallowing process. An association between malnutrition and dysphagia has been reported in the post-acute stroke phase. However, the causality link between the two remains unclear [14]. After motor stroke, predominantly in the paretic limb but not only, the size of muscle fibers typically decreases [15]. The muscle fiber type switches from II to I (fast to slow) progressively [16,17]. Moreover, as the muscle mass decreases after stroke, the proportion of intramuscular fat increases in the paretic limb [18,19]. Stroke-related sarcopenia and its combination with poor nutritional status appear highly interlinked, especially, in the older stroke population in which it may affect the stroke outcome. Therefore, methods that prevent muscle loss and improve muscle function after stroke are of great interest. This systematic review aims at identifying nutritional and pharmacological interventions which may prevent loss of muscle mass in the elderly after stroke.

## 2. Materials and Methods 

A systemic literature search was performed according to the Preferred Reporting Items for Systematic Review and Meta-Analysis Protocols (PRISMA-P) guidelines [20]. The information source was the MEDLINE electronic database (PubMed search platform, 1998 onwards). The initial search was performed in December 2018, with the last search performed on April 2 2019. The eligibility criteria were defined before performing the search and were as follows. Study design: Randomized controlled trials, controlled trials, prospective cohort studies, retrospective comparative cohort studies, case control studies and case-matched studies. Subject characteristics: Geriatric subject (mean age of groups > 60 years old), who were stroke survivors, who went through a neurorehabilitation program. Timing: Minimum 1 month follow up. Language: English. Based on the association of six blocks using Boolean operators, the search strategy was the following: [“Evidence” OR “Randomized controlled trial” OR “Cohort study” OR “Case-control study” OR “Qualitative research”] AND [“Neurological rehabilitation” OR "Stroke”] AND [“Elderly” OR “Aged” OR “Geriatric” OR “Frail”] AND [“Sarcopenia” OR “Muscle” OR “Muscle atrophy” OR “Muscle strength”] AND [“Occupational therapy” OR “Physical therapy” OR “Cognitive therapy” OR “Recreation therapy” OR “Nutrition” OR “Nutrition therapy” OR ”Intervention” OR “Hormonal replacement” OR “Drug”] OR [nutrition AND aged AND "stroke rehabilitation"]. One reviewer assessed the abstracts retrieved from the literature search. For abstracts fulfilling the major eligibility criteria (neurorehabilitation, stroke survivors), the full text was further screened to address the age of the subjects. Then, from all the papers that met the age criteria, those evaluating interventions based on nutritional or pharmacological strategies that affect muscle mass or muscle function were selected. The study selection process is depicted in the flow chart of Figure 1. A standardized scale for quality assessment was used to rate the selected studies. The checklist proposed by Downs and Black [21] was modified. Item number 27 was rated as 1 if a sample size calculation was reported by authors or 0 if not. Therefore, the maximum score was 28.

## 3. Results

Using the defined criteria, eight randomized and two cohort studies were selected for further review. However, one of the studies was excluded because it had been retracted by the editors because of concerns about data integrity and scientific misconduct [22,23]. From the nine remaining studies, see Table 1, four were multicenter studies [24,25,26,27]. Two interventions were commenced in the acute stroke phase [25,28] while others were including chronic stroke patients either during rehabilitation as inpatients [29,30] or outpatients [24]. Randomized control trials were categorized as unimodal when the methods included a single specific intervention in addition to a classical neurorehabilitation program or multimodal when the protocol combined different approaches. As in most studies in neurorehabilitation, they are not standardized; therefore, the functional outcomes measured in these studies could vary according to the intervention. 

### 3.1. Unimodal Interventions

Ha et al. showed that careful nutritional risk assessment (by estimating energy need and dietary intake) and an individual nutritional support (to cover estimated energy need) performed during the first seven days after admission for acute stroke could reduce the number of patients with a body weight loss ≥5% at three months (risk reduction of 15.7%, *p* = 0.055) [28]. While the intervention was associated with a stable protein intake across groups (0.8 vs. 0.7 g/kg, *p* = 0.065) the overall energy intake was 14.6% higher than that in the control group (1197 vs. 1045 kcal/day, *p* = 0.032). As a consequence, in the intervention group, there was a higher number of patients with improved handgrip strength (risk increase of 28.6%, *p* = 0.001). The mean handgrip strength was increased by 10.6% as compared to the control group (*p* = 0.002). Moreover, the patients reported a significant improvement in the quality of life in the mobility items of the EQ-5D questionnaire. All the outcomes of the study were measured three months after study entry. The only information provided on patient trajectory is the length of hospital stay, which was not significantly different between the groups. The authors did not provide data on the proportion of patients that went through a formal rehabilitation program after the acute stroke phase and their distribution in the study groups. Such data would have allowed for excluding potential bias in this study.

Rabadi et al. compared the effects of two commercially available nutritional supplements (standard: 127 kcal, 5 g proteins vs. intensive: 240 kcal, 11g proteins) on functional outcomes in a neurological rehabilitation setting in patients who presented with a minimal 2.5% weight loss within 14 days following stroke onset [31]. As compared to standard nutritional supplementation, upon discharge, patients in the intensive supplement arm of the study had a significantly greater motor functionality (motor items of the Functional Independence Measure (FIM scale) score of 24.25 vs. 16.7, *p* < 0.001). This beneficial effect on motor skills was also confirmed in the 2-minute and 6-minute walk tests in which patients from the intensive arm had a significantly greater improvement (2-minute: 101.6 vs. 43.98 feet, *p* < 0.001; 6-minute: 299.28 vs. 170.59 feet, *p* < 0.001). This effect allowed significantly more patients to return home (63 vs. 43%, *p* < 0.05). While the daily dietary intake was not monitored in this study, the intensive nutritional supplementation led to a non-significant higher weight gain (2.31 vs. 0.67 lbs, *p* = 0.37). 

Some epidemiological data support the effect of omega-3 fatty acids intake in the prevention of ischemic stroke [33]. Moreover, the plasmatic antioxidative activity, as well as antioxidant vitamin levels, seem to be associated with stroke lesion size and neurological outcomes. Therefore, the Nutristroke trial evaluated the supplementation with omega-3 fatty acids (0.5 mg/day) and/or antioxidant (vitamin C 240 mg/day, vitamin E 290 mg/day, beta-carotene 19 mg/day and polyphenols 150 mg/day) during rehabilitation of post-stroke patients [32]. This study failed to demonstrate a significant functional improvement in treated patients. There was, nevertheless, a trend toward a lower mortality rate at the one-year-follow-up point in subgroups of patients treated with antioxidants with or without omega-3 fatty acids (*p* = 0.06). However, this finding needs to be interpreted with caution because of the low number of patients per group.

Edaravone is a powerful intravenous free radical scavenging compound that has demonstrated some neuroprotective effects in many preclinical models [34,35]. It was approved in Japan in the acute stroke phase. Despite the fact that no placebo group was included in their study, Naritomi et al. reported that 10–14 days treatment with edaravone, as opposed to three days treatment, could significantly reduce muscle atrophy of a paretic leg three months after a stroke (difference of 4.7% atrophy in paretic leg, *p* < 0.01) [25]. Interestingly, an effect on walking speed was also reported and may indicate a functional effect related to this observation on muscle trophicity (97.9 ± 67.3 cm/second vs. 53.6 ± 54.8 cm/second, *p* < 0.05). 

Okamoto et al. tested the effect on thigh muscle of the intramuscular injections of the anabolic steroid, metenolone enanthate (ME) over a six-week timeframe in hemiplegic stroke patients [30]. Authors reported a significant increase in the cross-sectional area (CSA) of the thigh in both affected and unaffected legs. Despite a low number of treated patients, after six weeks, a significant increase in CSA was observed in the intervention group as compared to controls (13.4% vs. 3.3% increase in the paretic side and 14.5% vs. 5.2% in non-paretic side, *p* < 0.05). The authors did not provide any information on the sustainability of this effect after six weeks. Unexpectedly, this increase seemed to be greater in patients with a lower score for the motor items of the FIM scale. The authors suggest that this observation might be related to the fact that patients with lower motor FIM admitted to their rehabilitation ward are, typically, more inactive in the acute care hospital and, therefore, have more room for improvement.

### 3.2. Multimodal Interventions

Yoshimura et al. combined a 3 g of leucin 40% enriched amino acids supplement with low-intensity resistance training on top of a classical rehabilitation program [29]. Both groups received the resistance training, which consisted of a sit-to-stand exercise, from 10 up to 120 repetitions as the strength and durability improved during the protocol. The rehabilitation program was tailored to the patient’s need and included physical, occupational and speech therapy. After eight weeks, a greater improvement was observed in motor functionality (+9.2 points in FIM-motor score, 95%CI 1.5–15.8, *p* = 0.045) and handgrip strength (+3.8 kg, 95%CI 1.09–7.22, *p* = 0.002) in the intervention group. 

In an attempt to prevent falls after discharge from hospital in stroke patients at risk, Batchelor et al. evaluated a multimodal intervention [24]. The intervention group received an individualized exercise program, an educative program, the implementation of multiple fall and injury prevention strategies and adequate vitamin D and calcium supplementation. However, this study could not demonstrate a clinical benefit as the fall rate and other outcomes did not significantly differ between groups. 

### 3.3. Cohort Studies

In a retrospective multicenter cohort study including 192 geriatric stroke patients with prescribed rehabilitation, Kokura et al. evaluated the impact of energy intake during the first week after admission on the functional stroke outcomes [26]. The patient population was segregated into two subgroups according to their daily energy intake. The mean energy intake in the “energy sufficiency” group was 1206 ± 257 kcal/day, whereas it was 807 ± 394 kcal/day in the “energy shortage” group (*p* < 0.001). In a multivariate statistical analysis, energy sufficiency was significantly independently associated with total FIM gain (27 vs. 9; β = −0.166, 95%CI −7.295 to −0.175, *p* = 0.039) and motor-FIM gain (26 vs. 6; β = −0.205, 95%CI −7.191 to −1.027, *p* = 0.009). The absence of energy sufficiency was independently associated with the presence of complications (pneumonia, urinary tract infection or pressure ulcer) during hospitalization (OR 3.794, 95% CI 0.148–1.266, *p* = 0.017).

James et al. compared stroke patients who received tube feeding for nutritional support with those who did not in a post-stroke inpatient rehabilitation database [27]. This study highlights major variability in the use of tube feeding across centers. Nevertheless, in a regression analysis, tube feeding for 1–24% or ≥25% of the rehabilitation stay was significantly associated with increased total FIM (45.1 ± 12.6 and 33.9 ± 20.3, respectively, *p* = 0.002 and 0.005) and motor-FIM scores (36.6 ± 13.0 and 26.9 ± 16.8, respectively, *p* = 0.006 and *p* = 0.004) in severe strokes at discharge. Tube feeding for ≥25% of the rehabilitation stay also had a greater improvement in the severity of illness at discharge measured by the Comprehensive Severity Index (CSI) (16.8 ± 12.4; *p* = 0.001). Interestingly, the group exclusively fed by tube was associated with poor functional recovery, reflecting more post-stroke significant impairment. Despite the fact that energy intake was not reported in this study, it confirms the importance of early nutritional support in stroke rehabilitation.

## 4. Discussion

The literature reveals that interventions were heterogeneous and very limited in number in elderly post-stroke patients regarding muscle mass and muscle functions.

The two studies evaluating nutritional interventions demonstrated a significant improvement in muscle mass or handgrip strength [28,29]. Handgrip isometric strength is a validated quantitative marker of muscle function that is related to lower limb muscle function and calf cross-sectional muscle area [36]. It has been proposed as a valuable tool for the diagnosis of age-related sarcopenia [8]. The interventions tested by Yoshimura et al. resulted in an increased protein intake (>1.2 g/day), which is consistent with nutritional recommendations for the prevention of age-related sarcopenia [37]. Because stroke- and age-related sarcopenia share similar features [38], the generalization of these recommendations to older stroke patients seems to be confirmed by the identified studies. Interestingly, leucin-enriched amino acid supplementation may be less effective when not combined with exercise [39]. It means that such nutritional intervention is more effective for muscle mass when combined with resistance exercise as previously demonstrated in other settings [40]. The retrospective study by Kokura et al. confirmed that post-stroke elderly patients with insufficient energy intake have inferior motor and functional outcomes [26]. Nevertheless, the routine administration of nutritional supplements in a broad range of post-stroke patients did not provide beneficial effects on stroke outcomes [41]. However, the study by Rabadi et al., performed in selected undernourished patients (with evidence of weight loss after acute stroke), demonstrated that an increased dietary intake provided by nutritional supplements improved motor recovery and functionality [31]. Interestingly, in this study, more than 55% of the included patients had dysphagia, a condition that is strongly associated with malnutrition and higher morbidity and mortality rates [14,42]. James et al. demonstrated that nutritional support with tube feeding in rehabilitation is associated with greater motor improvement [27]. Taken together, these data support a systematic nutritional assessment and treatment of malnutrition in elderly stroke patients.

The consumption of higher relative levels of omega-3 fatty acids and antioxidants in the Mediterranean diet or through the consumption of fish is associated with a lower incidence of stroke and other cardiovascular events [43,44]. However, the Nutristroke trial failed to demonstrate significant effects of omega-3 fatty acids and antioxidant supplementation during stroke rehabilitation on patient functional status [32]. The mechanisms involved in the preventive effects of these dietary compounds on stroke occurrence may differ from those involved in stroke recovery and, therefore, this supplementation is not recommended.

Vitamin D supplementation has demonstrated a slight but significant effect on muscle strength particularly, in people aged 65 and over [45]. However, the study by Batchelor et al. failed to demonstrate the efficacy of a multimodal fall prevention intervention after stroke [24]. These findings are consistent with recent meta-analyses showing no effect of vitamin D supplementation on muscle health [46].

Anabolic steroids, especially testosterone, have been proposed to counteract age-related sarcopenia in older men. Testosterone replacement therapy is currently recommended to increase muscle mass and muscle function in older men only when serum testosterone levels are low [47]. Treatment for six weeks with the anabolic steroid metenolone enenthate during stroke rehabilitation increased the thigh muscle cross-sectional area [30]. However, this study could not demonstrate an effect on functional outcomes. Adverse events reported during the study included an increased fasting blood sugar level, hyperproteinemia or liver dysfunction. Anabolic steroids are classically associated with hepatic, metabolic or cardiovascular events [48,49]. Despite the fact that these observations were resolved at the end of the treatment, they confirm that the safety profile of such treatment should be carefully addressed in this specific patient population. As testosterone doesn’t seem to improve function in such a setting, and as their effect is reversible also for the beneficial effect on muscle mass [50], it doesn’t seem to be reasonable to propose this treatment after a stroke.

Edaravone was approved in 2001 in Japan for the treatment of ischemic stroke. The approval was based on the results of a phase-3 randomized, placebo-controlled, double-blinded multicenter study that demonstrated a significant improvement of stroke outcomes in treated patients (higher proportion of patient with lower disability measured by the modified Rankin scale) [51]. A recent retrospective observational study based on the Japanese stroke registry confirmed a potential benefit of this drug in the treatment of strokes [52]. Moreover, edaravone was recently approved in many countries for the treatment of amyotrophic lateral sclerosis (ALS) after a smaller decline of muscle strength in treated subjects was demonstrated [53]. The study by Naritomi et al. compared two treatment regimens with edaravone during the acute stroke phase [25]. Despite the fact that no placebo group was included in the study, longer treatment duration may be associated with a decreased muscle atrophy three months after stroke, suggesting a potential myoprotective effect of this drug. However, this effect needs to be confirmed in a larger patient population in randomized controlled trials. It would also be interesting to include, in future trials, some measures of muscle strength and function to confirm the benefit of this drug. There is currently not enough data to support its use. 

The present review has some limitations. The literature search was performed only in one database (MEDLINE). Using different sources may have resulted in additional references and would have possibly allowed the inclusion of unpublished data. The strengths of this work rely on its systematic methodology and its originality as an effort to study a specific patient population in a specific setting. 

## 5. Conclusions

The present systematic review demonstrates that little evidence supports interventions to prevent a decrease in muscle mass and function in elderly post-stroke patients. It tends to suggest that nutritional interventions recommended in the prevention of age-related sarcopenia, in association with a rehabilitation program, may be effective after stroke. However, this review highlights the need for future studies in this particular patient population as the number of older stroke patients with a residual disability is expected to continue growing in the years to come. 

## Figures and Tables

**Figure 1 nutrients-11-00881-f001:**
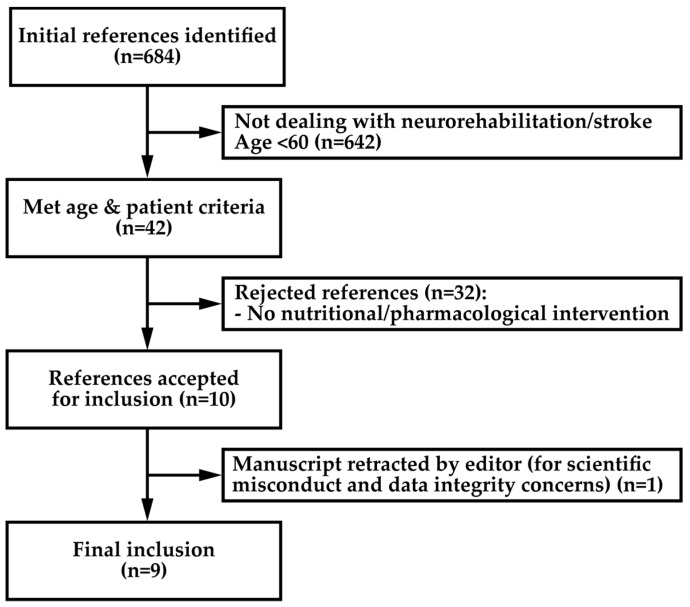
Flow chart of studies selection process.

**Table 1 nutrients-11-00881-t001:** Selected studies.

Studies		Population	Intervention/Hypothesis	Outcomes	Results	Major Limitations	Quality Score
***Unimodal RCT***	Ha et al. 2010 [28]	Post-stroke INT = 58/CO = 66	Individual nutritional support	Body weight at 3 months, QoL, handgrip	Significant reduction of patients with weight loss, improvement of QoL, increase handgrip strength	High dropout rate, single center, no evaluation of impact of rehabilitation	19/28
	Rabadi et al. 2008 [31]	Post-stroke with ≥2.5% weight loss INT = 51/CO = 51	Intensive nutritional supplement	FIM, length of stay, 2-/6-minute timed walk	Significant increase in motor-FIM gain, increased gain in walking performance, increased proportion of return to home	Single center, no placebo, no monitoring of dietary intake	22/28
	Garbagnati et al. 2009 [32]	Post ischemic stroke INT = 16/20/18CO = 18	Supplement with antioxidants or omega-3 FA or both	BI, RMI, CNS	Trend toward lower mortality rate	Single center, high dropout rate, small sample size	18/28
	Naritomi et al. 2010 [25]	Post-stroke with leg motor weakness INT = 21/CO = 20	10–14 days edaravone treatment	Femoral muscle volume, BRS, MWS	Significant decrease in muscle atrophy, improved leg function	Small sample size, lack of placebo group, open label	20/28
	Okamoto et al. 2011 [30]	Hemiplegic stroke patients INT = 15/CO = 11	6-weeks treatment with anabolic steroid	CSA of the bilateral thigh muscles, Motor-FIM, ADL score	Significant increase in bilateral thigh CSA	Small sample size	16/28
***Multimodal RCT***	Yoshimura et al. 2019 [29]	Post-stroke, sarcopenic, INT = 21/CO = 23	8-weeks administration of leucin-enriched amino acids + low intensity resistance training	FIM, SMI, handgrip	Significant improvement of motor FIM, SMI and handgrip	Small sample size, single center, association with exercise	24/28
	Batchelor et al. 2012 [24]	Post-stroke, high fall risk INT = 57/CO = 75	12-months fall prevention program (including calcium/ vitamin D supplementation	Fall rates, leg strength, FIM, gait speed, balance, fall risk	No significant difference	Subjects not blinded	22/28
***Cohort studies***	Kokura et al. 2018 [26]	Post-stroke, rehabilitation prescribed *N* = 192	Impact of energy intake during first week	FIM, complications	Energy intake during week 1 significantly affects motor outcomes	Retrospective, high exclusion rate, energy intake not monitored after week 1	22/28
	James et al. 2005 [27]	Moderate and severe stroke *N* = 919	Impact of tube feeding	FIM	Greater functional improvement in tube-fed severe stroke survivors	Observational, no causal effect is demonstrated	17/28

RCT: Randomized control trial, INT: Intervention group, CO: Control group, QoL: Quality of life, FIM: Functional Independence Measure, FA: Fatty Acids, BI: Barthel Index, RMI: Rivermead Mobility Index, CNS: Canadian Neurological Scale, CSA: Cross-sectional area, ADL: Activities of daily life, BRS: Brunnstrom Recovery Stage, MWS: Maximum walking speed, SMI: Skeletal Muscle Index.

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
