# Peer review of "How to Prevent Loss of Muscle Mass and Strength among Older People in Neuro-Rehabilitation?"

_nutrients, 2019, doi:10.3390/nu11040881_

Reviewer 1 Report

The reviewers attempt to review evidence of nutritional interventions that may prevent loss of muscle in older stroke patients.  However, upon evaluation of included studies, the authors appear to be missing several key articles calling into question the validity of their search criteria.  For example, the NutriStroke trial (Cerebrovasc Dis. 2009;27(4):375-83.) is not discussed.

It is also not clear as to why the neuro-rehabilitation requirement was chosen for eligibility as some of the articles do not appear to clarify the role of rehab (i.e. H a et al.). If this category is removed, there are several other studies worth including (i.e. VITATOPS, VISP trial).

Further, I am not sure why a limit of 20 years was chosen, as there were other older articles that examined outcomes associated with sarcopenia, such as strength, that I believe are worth mentioning (Stroke. 1997 Apr;28(4):736-9).

The authors may also want to consider the inclusion of bone-related outcomes since this is so closely related to sarcopenia.

Author Response

Please see attached PDF file

Reviewer 2 Report

Lathuiliere et al. carried out a systematic literature review to identify the evidence for nutritional or pharmacological interventions which might prevent the loss of muscle mass in the elderly after stroke. This manuscript was well-described and would be a helpful review for stroke-induced decreased in muscle mass and force.

In abstract, because the authors include the interventional study using anabolic steroid, the authors should describe “nutritional and pharmacological intervention” in line 14.

In line 46, “switches from II to I (slow to fast)” might be “switches from II to I (fast to slow)”.

In line 59, “geriatric subject (mean age of groups > 60 years old), stroke survivors” should be “geriatric subject (mean age of groups > 60 years old) who were stroke survivors”. It might be not clear “geriatric subject and stroke survivors” or “geriatric subject or stroke survivors”.

In line 76, (Table 1) should move to line 78 (after “the five remaining studies), because Table 1 included only 5 literature.

In Table 1, “Ha” should be “Ha et al”. Other 4 literature should be revised in a similar manner.

In line 89, “seven” would be “7”.

In line 99, “do” should be “did”.

In 109, need the explanation for “FIM”.

In line 117, “p>0.01” might be “p<0.01”.< span="">

In line 143, need the reference for “stroke- and age-related sarcopenia shares similar features”.

In line 186, “prevent muscle mass and function” should be “prevent the decrease in muscle mass and function”.

Author Response

Please see attached PDF file

Round  2

Reviewer 1 Report

With the new search criteria, the authors have addressed the reviewers main concerns.

Author Response

Thank you

Reviewer 2 Report

The authors fully answered my questions.

Author Response

Thank you